# Selected Fungal Natural Products with Antimicrobial Properties

**DOI:** 10.3390/molecules25040911

**Published:** 2020-02-18

**Authors:** Dorota Jakubczyk, Francois Dussart

**Affiliations:** 1Institute of Bioorganic Chemistry, Polish Academy of Sciences, 61-704 Poznań, Poland; 2Department of Agriculture, Horticulture and Engineering Science, Scotland’s Rural College (SRUC), Edinburgh EH9 3JG, UK; Francois.Dussart@sruc.ac.uk

**Keywords:** antimicrobial properties, antimicrobial resistance, biosynthesis, ergot alkaloids, fungal metabolites, natural products, peptides, polyketides, terpenoids

## Abstract

Fungal natural products and their effects have been known to humankind for hundreds of years. For example, toxic ergot alkaloids produced by filamentous fungi growing on rye poisoned thousands of people and livestock throughout the Middle Ages. However, their later medicinal applications, followed by the discovery of the first class of antibiotics, penicillins and other drugs of fungal origin, such as peptidic natural products, terpenoids or polyketides, have altered the historically negative reputation of fungal “toxins”. The development of new antimicrobial drugs is currently a major global challenge, mainly due to antimicrobial resistance phenomena. Therefore, the structures, biosynthesis and antimicrobial activity of selected fungal natural products are described here.

## 1. Introduction

Natural products (NPs) are a very rich source of antimicrobial drugs. They constitute more than two-thirds of clinically used antibiotics and half of anticancer drugs [1]. Molecules biosynthesised by fungi are a diverse and useful group of NPs. Plant endophytic and pathogenic fungi produce many secondary metabolites that play important roles in virulence and competition against other microbes. Due to their broad-spectrum activity, some of these NPs can also exhibit high biocidal activity against human pathogenic microbes. In recent years, marine fungi have emerged as a novel source of fungal NPs and may be a potential game changer in drug discovery; however, marine fungi still constitute an underrepresented resource of diverse NPs.

Fungal NPs have an important place in human history. For instance, ergot alkaloids (EAs) which are produced by the filamentous fungi of the genus *Claviceps*, have been referenced in ancient historical texts. References to grain diseases have been found in the Bible, in the Old Testament (850–550 BC). In the Middle Ages, the first reported ergotism epidemic was recorded in 944–1000 AD when almost half the population of the Aquitaine region of France (about 60,000 people) died of ergot poisoning [2,3]. The gangrenous form of the disease (medically known under the name Ergotismus gangraenosus) was commonly known as ‘‘ergotism’’, ‘‘holy fire’’ or ‘‘St. Anthony’s fire’’. Symptoms include delirium, hallucinations, muscle spasms, convulsions and gangrene of the limbs. The gruesome history of ergots has overshadowed their beneficial medicinal properties. However, the use of ergots as medicinal compounds was first documented in 1582, as they were administered for ‘‘quickening childbirth’’. Further research and screening of ergot analogues for oxytocic drugs that stimulate uterine contractions to hasten childbirth resulted, in 1938, in the synthesis of lysergic acid diethylamide (LSD) **5** (Figure 1B), a hallucinogenic compound that has become infamous for its use as an illicit “recreational drug” [4]. Currently, EAs are the inspiration for numerous semi-synthetic derivatives, such as cabergoline **6** or ergotamine **8** (Figure 1B,D, respectively) that have been applied in a wide range of medicinal treatments, such as the treatment of migraines, Parkinson’s disease, reduction of tumour growth, and other lesser-known synergistic antimicrobial activities.

Another example of the historical importance of fungal NPs was the discovery of penicillin **32** by Sir Alexander Fleming in 1928. The identification of this first antibiotic compound from the mould *Penicillium notatum* was a breakthrough and a revolution in drug discovery [5]. Ever since penicillin was discovered, a completely new era of chemotherapy started, thereby changing the quality of human life. The importance of this life-saving discovery cannot be understated, as penicillin was used to cure countless people of bacterial diseases. The development of penicillin initiated the golden era of natural antibiotics. The search for bioactive NPs led to the discovery of a multitude of antibacterial compounds, many of which were isolated from *Streptomyces* species [6]. The genus *Streptomyces* regroups species of actinobacteria that share morphological traits with fungi, such as filamentous hyphae and spore production. To date, over 350 agents derived from diverse chemical classes of metabolites isolated from *Streptomyces* have reached the world market as antimicrobial compounds [7]. These bioactive agents include NPs, semi-synthetic antibiotics and synthetic compounds [7,8]. However, the wide use of antibiotics has resulted in the development of resistant microbes due to the evolutionary selective pressure driven by antibiotics [9]. The number of effective therapeutics against life-threatening bacterial and fungal infections has fallen dramatically because of emerging multidrug-resistant (MDR) pathogens.

Antimicrobial resistance (AMR) is a major concern of modern medicine and it has now become one of the key research areas of the European Union Commission [10]. AMR occurs when microbes, such as bacteria, fungi, viruses and parasites acquire resistance to one or more drugs. Drug resistance is the biggest obstacle to success during the treatment of infectious diseases, and has been observed following the introduction of numerous antimicrobial agents into clinical practice. It is difficult to quantify the global human burden posed by AMR but, in the European Union alone, 25,000 people die every year due to drug-resistant bacterial infections [10]. Resistance to antimicrobial compounds also has a major impact on food production worldwide. Since the Green Revolution (ca. 1950–1970), food production and agriculture has been reliant on chemical input to control pathogenic microbes, either in animal or plant production [11]. However, the overuse of these chemicals, combined with the lack of diversity in their modes of action, has driven the appearance of resistance to these compounds. As a result, the control of pathogenic microbes has become increasingly difficult in the past few decades, contributing to the increased volatility of food production and food insecurity. Drug resistance is driving the constant need for new drug discoveries. To manage the risk of resistance to antimicrobial compounds arising, efforts are being made to better understand the mechanisms underlying host-microbe interactions, pathogen population evolution and drug modes of action. Herein, the structures, biosynthesis and antimicrobial activities of selected natural products derived from fungi are presented.

## 2. Selected Examples of Antimicrobial Natural Products from Fungi

### 2.1. Ergot Alkaloids: Fungal Natural Products Derived from Amino Acids

All naturally occurring EAs share a common tetracyclic scaffold, the so-called ‘‘ergoline scaffold”, derived from L-tryptophan. EAs are divided into three major classes based on the substituents decorating this scaffold: clavines (festuclavine and agroclavine derivatives), simple lysergic acid derivatives and ergopeptides (Figure 1A,B,D, accordingly) [12]. Clavines include the partially or fully saturated ring species D, such as agroclavine **1**, festuclavine **2** or lysergol **3** (Figure 1A). Simple lysergic acid derivatives consist of the basic D-lysergic acid structure as an alkyl amide (Figure 1B), and ergopeptides also based on D-lysergic acid and a cyclic tripeptide moiety (Figure 1D). Cycloclavine **7** is a newly characterised ergot alkaloid which has been reproduced *in vitro*, and has an unusual ring system, where ring D has been transformed into a new five- and three-membered ring fusion [13].

EAs are produced by fungi occupying distinct ecological niches. *Clavicipitaceous* species, such as *Claviceps purpurea* and *Neotyphodium lolii* from the order *Eurotiales* are plant pathogenic and symbiotic fungi, respectively. *Aspergillus fumigatus* from the same order, *Eurotiales*, is an opportunistic pathogen of mammals which also produces EAs, such as festuclavine **2** [15,16]. Cycloclavine **7** is biosynthesised in nature by *Aspergillus japonicus*, which is frequently responsible for the post-harvest decay of fresh fruit (apples, pears, peaches, citrus, grapes, figs, strawberries, tomatoes or melons) and some vegetables (especially onions, garlic, and yams) [17].

The biosynthetic pathways of EAs have been well studied and are described in depth elsewhere [14,18]; however, a brief overview of EAs biosynthesis is given here. First, the prenylation of L-tryptophan by dimethylallyl pyrophosphate (DMAPP) yields 4-(γ,γ-dimethylallyl)tryptophan (DMAT) and is followed by the *N*-methylation of DMAT to 4-dimethyl-L-abrine (*N*-Me-DMAT). Subsequently, a series of successive oxidation steps catalyses the intramolecular cyclization of the prenyl and indole moieties to form ring C in tricyclic chanoclavine-I, which, in turn, is oxidised to form chanoclavine-I-aldehyde. At this branch point, chanoclavine-I-aldehyde undergoes intramolecular cyclization to form either ring D of tetracyclic agroclavine **1** (*C. purpurea, N. lolii*) or festuclavine **2** (*A. fumigatus*). Subsequently festuclavine **2** is further biotransformed into fumigaclavines. The new branch of this pathway is an unusual oxidation of the cyclic iminium form of chanoclavine-I-aldehyde catalysed by non-heme iron and α-ketoglutarate dependent oxidase EasH, to yield a unique cyclopropyl ring moiety which is fused to a five-membered ring, in cycloclavine **7**. The structure of EasH and possible mechanism of cycloclavine **7** formation has been published recently [19,20].

Ergot-derived medicines, such as ergometrine **4**, were used to facilitate obstetric deliveries or to treat postpartum haemorrhage. The high bioactivity of EAs is correlated with the ability of these compounds to act as agonists or antagonists toward neuroreceptors. Although some EAs, such as the hallucinogenic compound LSD **5** have been used as recreational drugs, most EAs were associated with medicinal applications, including treatments against migraine and tumour (ergotamine **8**), Parkinson’s disease or restless leg syndrome (cabergoline **6**; Figure 1B). However, their synergistic antimicrobial activity is a less commonly known fact. Lysergol **3** is a synthetic EA that exhibits a synergistic antibiotic pharmaceutical activity as a bioactive enhancer and a bioavailability facilitator for broad-spectrum antibiotics. This property facilitates the absorption of antibiotics across the cell membrane in animal cells resulting in increased action against Gram-positive and -negative bacteria [12]. With the recommended dosage of lysergol of 10 μg/mL, the improved activity of antimicrobial effect is in the range of 2–12 folds, against a wide spectrum of both Gram-positive and -negative bacteria including *Escherichia. coli*, *Bacillus subtilis*, *Mycobacterium smegmatis* and other similar microorganisms.

### 2.2. Fungal Polyketides

The polyketide pathway constitutes one of the major biosynthetic pathways leading to the production of fungal NPs. Polyketides are polymers synthesised from simple carboxylic acid derivatives (e.g., acetyl-CoA, malonyl-CoA, and methylmalonyl-CoA) into linear chains by iterative Claisen condensation, followed, in some cases, by reductive modification of the resulting β-keto groups. These compounds are synthesised in fungi (and other organisms) by enzymes called polyketide synthases (PKSs). Polyketides are extremely diverse and include compounds such as polyesters, polyphenols, macrolides (macrocyclic esters), polyenes and enediynes.

Strobilurins are an important group of polyketide-derived fungal NPs which have yielded one of the major classes of fungicides currently in use to protect agricultural crops from fungal diseases. The discovery of these compounds occurred after the observation that *Strobilurus tenacellus* and *Oudemansiella mucida*, two agaricomycetes growing on decaying wood in European forests, were able to defend themselves against other fungi. Their antifungal activity was associated with the production of the compounds strobilurin A **9** and oudemansin A **10** in *S. tenacellus* and *O. mucida*, respectively (Figure 2) [21,22]. These two compounds inhibit the transfer of electrons between complexes II and III of the electron transport chain in the mitochondria, resulting in impaired cell respiration and ATP synthesis [23]. Despite strobilurin A **9** and oudemansin A **10** exhibiting high antifungal activity, these NPs are quickly degraded by light, rendering them unsuitable for use in crop protection. Many attempts were made to modify the chemical structures of natural strobilurins to increase photo-stability while maintaining antifungal activity [24]. After several years of research, azoxystrobin **11** was synthesised and became the first photo-stable strobilurin-derived active ingredient registered for use in crop protection (Figure 2) [25], paving the way for the synthesis of a multitude of fungicides belonging to the quinone outside inhibitor (QoI) class of fungicides. However, resistance to QoI fungicides arose after a few years of use in fields. The single point mutation which confers resistance to QoI leads to the substitution of the amino acid glycine for alanine at position 143 (G143A). This mutation is now widespread in many fungal species, including *Zymoseptoria tritici*, *Botrytis cinerea* and *Cercospora beticola*, the agents responsible for Septoria leaf blotch in wheat, Botrytis grey mould and Cercospora leaf spot in beets, respectively [26,27,28]. Despite numerous resistance issues, QoIs are still used to control some of the most devastating rust fungi, such as *Puccinia striiformis* and *Phakopsora pachyrhizi*, the causative agents of the yellow rust of cereals and soybean rust diseases, respectively [29].

Some of the polyketide NPs synthesised by pathogenic fungi exhibit dimeric structures. Such is the case of lesser-known compounds produced by the *Torrubiella* species. The *Torrubiella* species are arthropod-pathogenic fungi that parasitise spiders, scale-insects and hoppers and are known to synthesise derivatives of uredinorubellin I **12** and II **13** (Figure 2). These compounds exhibited photodynamic activity, influencing cell viability in three mammalian cell lines, such as HIG82, HT29 and J774A.1, as well as antibacterial activity against *Staphylococcus aureus* [30]. The genus *Torrubiella* which belongs to the *Clavicipitaceae* family, is related to the genus *Ramularia* which also contains fungi that produce polyketide NPs with antimicrobial activity.

*Ramularia collo-cygni* is an ascomycete fungus responsible for the important plant disease Ramularia leaf spot (RLS) [31]. RLS is primarily a disease of barley but the fungus can infect other grain crops, such as wheat and oats as well as wild grasses. *R. collo-cygni* produces a range of secondary metabolites, including rubellin anthraquinones **14**–**17** (Figure 2). Rubellins are non-host-specific phytotoxins with photodynamic properties [32]. Miethbauer et al. showed that rubellins are biosynthesised via a polyketide pathway, by demonstrating the incorporation of both [1-^13^C]-acetate and [2-^13^C]-acetate into the rubellins during their formation [33]. McGrann and co-workers have recently sequenced and analysed the genome of *R. collo-cygni* and found that it contains the genetic architecture to synthesise a wide range of secondary metabolites, including rubellins [34]. In a later study, it was suggested that the co-expression of genes coding for PKSs and hybrid PKS/nonribosomal peptide synthetases (NRPSs) may be associated with the competitive colonisation of the host plant and early symptom development [35]. However, no exact determination of the biosynthetic pathway of rubellins has been elucidated yet and the role of these metabolites remains unclear. Despite the phytotoxic properties of rubellins, these compounds may have potential pharmaceutical applications. Miethbauer et al. have observed initial activities against Gram-positive bacteria, including MDR strains, such as *B. subtilis* (ATCC) 6633, *S. aureus* (SG) 511, *S. aureus* 134/94 (MRSA), *Enterococcus faecalis* 1528 (VRE) or *Mycobacterium vaccae* (IMET) 10670 [36]. Rubellins also exhibit antimicrobial, antiproliferative, cytotoxic and tau aggregation inhibitory activity *in vitro* tests [36]. Minimal inhibitory concentrations (MICs) were determined with and without illumination, showing a light-dependent increase in the antibacterial activity of compounds **14**–**17** (except against *M. vaccae*), with rubellin D **17** being the most active.

PKSs are not only involved in the biosynthesis of anthraquinone derivatives, such as rubellins, but also that of other dimers, such as viriditoxin **18** or lindgomycin **19** (Figure 2). Viriditoxin **18** belongs to the group of xanthoradones produced by *Penicillium radicum* FKI-3765-2. Xanthoradones exhibit activity against methicillin-resistant *S. aureus* (MRSA) by inhibiting FtsZ, the bacterial tubulin homolog which is crucial in septum formation [37].

Lindgomycin **19**, an unusual antibiotic polyketide, contains two distinct structural domains, a bicyclic hydrocarbon and a tetramic acid that are connected by a carbonyl functional group. Naturally occurring tetramic acid derivatives, originating from a variety of marine and terrestrial fungi (Arctic fungus of the *Lindgomycetaceae* family), have attracted great interest due to the breadth of the spectrum of their biological activities, as well as their challenging structural complexity [38,39]. The majority of the compounds isolated to date have exhibited mostly antibiotic or antiviral activity. Lindgomycin **19** revealed good antibiotic activity against a number of Gram-positive bacteria (IC_50_: 2–6 μM), as well as the yeast *Candida albicans* and the plant pathogenic fungus *Z. tritici* (IC_50_: 5–10 μM) [40]. This compound also showed antibiotic activity against an MRSA strain with IC_50_ values of 5.1 μM. Another example of polyketide-derived NP from marine fungi is corollosporine **20** (Figure 2). Corollosporine **20** is an antibacterial phthalide derivative produced by the fungus *Corollospora maritima* which was isolated from driftwood found near the island of Heligoland, Germany [41].

### 2.3. Peptidic Fungal Natural Products

Two distinct pathways in fungi are responsible for the production of peptidic NPs. Enzymes in the nonribosomal peptide (NRP) pathway produce the majority of peptide metabolites. These are highly specific, multimodular enzymes called NRPSs, which utilise both proteinogenic and non-proteinogenic amino acids to synthesise the peptidic backbones. The genes encoding these enzymes are usually located within a biosynthetic cluster, comprising several genes that are co-regulated. The other pathway is that of ribosomally synthesised and post-translationally modified peptides (RiPPs); very large peptidic NPs with molecular weights typically around 1000 Da are synthesised through this pathway [42]. In this section we review selected groups of peptidic fungal NPs which have yielded valuable compounds with regards to cytotoxic properties and antibiotic drug discovery.

#### 2.3.1. Ribosomally Synthesised and Post-Translationally Modified Peptides (RiPPs)

Although many RiPPs isolated from bacteria exhibit antibiotic activity [43], at the time of writing only a limited number of fungal RiPPs have been identified. The main groups of fungal RiPPs were isolated from agaricomycetes in the genus *Amanita* and belong to the amatoxin and phallotoxin families [44]. Amatoxins and phallotoxins are highly toxic compounds exhibiting cyclic octo- and hepta-peptide structures, respectively [45]. Despite the cytotoxicity of these metabolites, amatoxins and phallotoxins have yielded valuable compounds, although admittedly not antibiotics. Amanitins **21**–**24** (Figure 3), which belong to the amatoxin family, are potent inhibitors of RNA polymerase II and were investigated as potential anti-cancer drugs. Amanitin derivatives suppressed the multiplication of several cancer cell lines, including pancreatic, colorectal, and breast cancer cells [46]. The ability of phallotoxins, such as phalloidin **25**, to bind actin filaments with a high affinity was used extensively to study cell biology, and phalloidin staining is still considered a gold standard for actin localisation in cells [47,48]. Ustiloxins **26**–**31** (Figure 3) were not originally identified as RiPPs when they were first isolated from the rice pathogen *Ustilaginoidea virens*, but the elucidation of their biosynthetic pathway has led to the reclassification of these metabolites as RiPPs [49]. Ustiloxins inhibit the assembly of microtubules during cell division, resulting in antimitotic properties and cytotoxicity against various cancer cells lines, including stomach, lung, breast, colon and kidney cancer cells [50]. Therefore, ustiloxins are of interest in the development of anti-cancer drugs.

#### 2.3.2. Nonribosomal Peptide Natural Products

##### β-lactam Antibiotics

The discovery of the β-lactam class of antibiotics is, arguably, one of the greatest advancements of modern medicine, as it has given rise to a myriad of bioactive molecules. Compounds belonging to the β-lactam family all contain a four-member cyclic amide group. The two major families of products in the β-lactams are penicillins and cephalosporins, originally isolated from the *Penicillium* and *Acremonium* species, respectively [51,52]. The initial step in the biosynthesis of penicillin and cephalosporin requires the condensation of the three amino acids L-α-aminoadipate, L-cysteine and L-valine. The reaction is catalysed by the NRPS δ-(L-α-aminoadipyl)-L-cysteinyl-D-valine (ACV) synthetase [53]. Subsequently, ACV is transformed into isopenicillin N, which serves as a starting compound for either penicillin **32** or cephalosporin **33**-derivative synthesis (Figure 4). β-lactams are the longest-serving antibiotics used in modern medicine and act against a broad spectrum of bacteria, including the pathogenic *Streptococcus*, *Staphylococcus*, *Enterococcus*, *Clostrodium* and *Treponema* species [54] by inhibiting the synthesis of bacterial cell wall peptidoglycans [55]. However, due to over-usage of β-lactams, many cases of resistance have been reported resulting in difficulties in controlling certain bacterial strains, including MRSA [56,57,58]. One of the most-studied mechanisms responsible for microbial resistance to antibiotics is the acquisition of genes encoding for β-lactamase enzymes by bacterial strains [59]. These genes, which can be integrated in the bacterial genome through horizontal gene transfer events, code for enzymes that are able to hydrolyse the β-lactam ring. The loss of the β-lactam ring is known to result in antibiotics inactivation. In addition to microbial resistance to antibiotics, cases of allergies to β-lactams have arisen, with about 10% of the population exhibiting allergic reactions to penicillins, rendering the control of bacterial diseases even more difficult [60]. Although only 1% of the population suffers from cephalosporin allergies, and allergies to other families of β-lactam are even less frequent, allergies are a major concern in clinical use. They are a particularly important consideration, as the ability to use different families of antibiotics including several β-lactams is crucial to prevent the development of multi-drug resistance [61,62].

##### Depsipeptides

Depsipeptides are peptides in which some amide groups are replaced by ester groups, resulting in lower susceptibility to proteolytic degradation [63]. Fungal depsipeptides have recently been reviewed at length [64], therefore, this section will focus on a few selected depsipeptides. Depsipeptides have been isolated from many ascomycete fungi, including plant pathogenic species in the genera *Fusarium*, *Alternaria*, *Calonectria* and *Cochliobolus* [65] as well as entomopathogenic fungal species, such as *Cordyceps cardinalis* and *Ophiocordyceps communis* [66,67]. Many depsipeptides exhibit both antimicrobial and insecticidal activity, and have, therefore, attracted attention for potential use in the development of new drug and crop protection products. Beauvericin **34** (Figure 4) is a family of natural products that were first isolated from *Beauveria bassiana*, the agent responsible for the white muscardine disease of arthropods. Many beauvericin derivatives have been identified and named alphabetically from A to J (including three G and H derivatives) and almost all of them exhibit biological activity including antimicrobial, antiviral and insecticidal activity; they can also inhibit cancer cell migration and be toxic to cancer cells [68,69,70,71,72]. The cytotoxicity of beauvericin is associated with the ability of this metabolite to form calcium channels in cell membranes, thus changing the intracellular cation levels and resulting in apoptosis, a form of programmed cell death [73,74]. Beauvericin was also reported to act as a potentiator of azole fungicides, resulting in reduced MIC against various strains of *C. albicans*, including azole-resistant strains [75]. Beauveriolide I **35**, which was isolated from the entomopathogenic fungus *Cordyceps militaris*, exhibits anti-aging activity in the baker’s yeast *Saccharomyces cerevisiae* [76], as treatment with beauveriolide I **35** extends the chronological lifespan of the yeast. The chronological lifespan of yeast, which is commonly used as a model for the ageing of cells, corresponds to the amount of time a yeast cell stays alive in a non-dividing state [77]. The many biological properties of fungal depsispeptides and the potential they represent for medicinal use have led to a reclassification of these compounds from mycotoxins to drug candidates [78,79].

##### Piperazines

Piperazines are marked by the presence of a six-membered ring containing two nitrogen atoms in opposite positions, and are the starting point for the biosynthesis of compounds in the diketopiperazine and epipolythiodioxopiperazine (ETP) families. Roquefortine C **36** (Figure 4) is one of the most famous and well-studied fungal diketopiperazines. It has been isolated from several *Penicillium* species, including *P. roqueforti*, the fungus responsible for the blue veins in some cheeses, such as Roquefort, Gorgonzola or Stilton [80]. The role and impact of this secondary metabolite on human and animal health has been subject to debate for many years. Several authors reported acute toxicity from roquefortine C in mice and dogs [81,82,83], and postulated that, as roquefortine is consistently present in silages, it posed a potential threat to cattle [84,85]. However, a consensus has emerged that doses found in human and animal diets are far below the level required for roquefortine C toxicity [86,87]. Interestingly, diketopiperazines play an important role in communication between fungi. The production of a set of two diketopiperazines by *Epichloë typhnia*, an endophyte of timothy (*Phleum pratense*), induces the production of the antifungal perylenequinone metabolite phleichrome by *Cladosporium phlei*, a pathogen of timothy [88]. Gliotoxin **37** (Figure 4) is an ETP produced by the opportunistic fungus *A. fumigatus*, which is responsible for aspergillosis in immunodeficient patients and plays a crucial role as a virulence factor during host colonisation [89]. Gliotoxin **37** exhibits immunosuppressive activity by inhibiting the activation of certain transcription factors involved in the mounting of the immune response, including lymphocyte B and T activation [90]. Although gliotoxin demonstrates potent antifungal activity against *C. albicans* and several *Aspergillus species* [91], its high toxicity and adverse effect on human health made it unsuitable for use as a potential drug.

##### Lipopetides

Several fungal species are known to produce secondary metabolites belonging to the lipopeptide class of NPs. Lipopeptides are molecules synthesised via the NRP pathway and consist of a lipid backbone fused to a peptidic moiety. The echinocandin family, which regroups lipidated cyclic hexapeptides, has been an important source of antifungal agents since the turn of the century. Echinocandin B **38** (Figure 4) was first isolated from *Aspergillus nidulans* in 1974 [92] but has since been reported in other *Aspergillus* species of the *Nidulantes* section [93]. Echinocandins A, B, C, D, and H are highly potent antifungal agents that act by inhibiting β-(1,3)-glucan synthase, the enzyme responsible for the biosynthesis of β-(1,3)-glucan, one of the main components of the fungal cell wall [94]. Within the echinocandin family, pneumocandins are also important antifungal agents. Pneumocandin B_0_
**39** (Figure 4) was first isolated from the saprophytic fungus *Glarea lozoyensis* [95] and exhibited high antifungal activity against *C. albicans* and *Pneumocystis carinii*, the agents responsible for candidiasis and pneumonia diseases [96]. Pneumocandin B_0_
**39** is now used as the starting compound in the production of the semi-synthetic antifungal drug caspofungin acetate; pneumocandin B_0_ is chemically modified to increase its molecule stability and bioactivity [97]. Cryptocandin **40** (Figure 4) is another powerful antifungal agent that also belongs to the lipopetide class of NPs, and has also attracted attention for its potential use as a medicinal drug. Cryptocandin **40** was isolated from *Cryptosporiopsis quercina*, a common fungal endophyte of hardwood species [98]. Cryptocandin **40** has shown antifungal activity against a broad range of fungal species, including the agent responsible for skin mycosis, *Tricophyton rubrum*, as well as the plant pathogens *Sclerotinia sclerotiorum* and *B. cinerea*, responsible for white and grey mould, respectively [99].

##### Other Peptides

Fungi have also yielded several other peptide metabolites of interest. One of the most important is, arguably, cyclosporine **41** (Figure 4). This cyclic undecapeptide was first isolated from *Toplypocladium inflatum*, an entomopathogenic fungus, and is synthesised by the NRPS cyclosporine synthetase [100]. Although cyclosporine **41** exhibits broad spectrum antifungal activity [101] its most valuable contribution is as an immunosuppressive drug administrated after organ transplant to avoid rejection. Cyclosporine **41** prevents the mounting of an immune response by inhibiting the activity of calcineurin, a calcium/calmodulin-dependent serine threonine phosphatase which activates transcription factors, therefore regulating the expression of cytokine genes involved in immunity [102].

Peptaibols are linear peptide fungal NPs composed of a short chain of amino acids, generally shorter than 20 residues. Several peptaibols exhibit promising antibiotic activity against a broad spectrum of pathogens. A set of peptaibols isolated from *Trichoderma reesei*, a fungus used industrially to produce cellulase enzymes, are extremely potent growth inhibitors of several fungal species, including the plant pathogens *Alternaria alternata*, *Phoma cucurbitaceum*, *Fusarium* spp., as well as the human pathogen *A. fumigatus* [103]. Similarly, trichokonin VI, a peptaibol isolated from *Trichoderma pseudokoningii*, triggers programmed cell death in a broad range of fungal phytopathogens, including *B. cinerea*, *F. oxysporum*, *Aschochyta citrullina* and the oomycete *Phytophtora parasitica*, resulting in growth inhibition [104]. Tulasporins **42**–**45** (Figure 4) are a group of four 19-mer peptaibols isolated from *Sepedonium tulasneanum*, a mycoparasitic fungus. All four tulasporins exhibit antifungal activity against *B. cinerea* and *P. infestans*, the pathogens responsible for Botrytis grey mould and potato late blight, respectively [105]. In addition to inhibiting the growth of many fungal plant pathogens, some peptaibols also have the ability to induce the expression of defence-related genes in plants, therefore conferring increased resistance to pathogens. Application of an 18-mer peptaibol produced by *T. virens* induces the expression of genes encoding key enzymes in different defence signalling pathways, with up-regulation of hydroxyperoxide lyase and phenylalanine ammonia lyase in the oxylipin and salicylic acid pathways, respectively, which highlights the potential of these compounds to elicit plant immune response [106].

### 2.4. Terpenoid Compounds

Terpenoids are a large class of natural products related to terpenes, and are made of condensed isoprene units. Terpenoids are often considered mostly as plant NPs; however, fungal species are also known to produce terpene-derived metabolites. Periconicin A **46** and B **47** (Figure 5) are secondary metabolites belonging to the diterpene family, which comprises compounds with four isoprene units. Both periconicin A and B exhibit antibacterial activity against several bacterial human pathogens, such as *S. aureus* and *Salmonella typhimurium* [107]. Similarly, enfumafungin **48** and ergokonin A **49** (Figure 5) are acidic terpenoids isolated from *Hormonema* and *Trichoderma* species, and which demonstrate antibiotic activity against several bacterial, yeast and fungal species, including *B. subtilis*, *Cryptococcus neoformans*, *C. albicans* and *A. fumigatus* [108,109]. The broad-spectrum activity of these two compounds was linked with their action as inhibitors of glucan biosynthesis [110]. The majority of bioactive fungal terpenes and terpenoids discovered in the past decade were isolated from marine fungi and fungi associated with algae [111]. For instance, peribysins **50**–**56** (Figure 5) are sesquiterpenes which were isolated from the dothideomycete *Periconia byssoides*, and are able to inhibit the development of leukemia cancer cells even at low concentrations, making compounds in this family potential anti-cancer drugs [112]. Similarly, the sesquiterpenoid insulicolide A **57**, which was isolated from the marine *Aspergillus ochraceus* Jcma1F17 strain, is a promising natural product that showed biological activity against viruses, such as the influenza virus H3N2 and inhibited the growth of diverse human cancer cell lines [113]. The screening of marine fungi for natural products with potential pharmaceutical applications has only recently begun, but has the potential to yield several new drugs.

## 3. Concluding Remarks

The production of antimicrobial compounds by fungal species has been suspected and utilised by humankind for centuries. For instance, healers in Egypt ca. 2650 BC were known to use mouldy bread to treat superficial wounds; whereas, in central Asia, clergymen used a mouldy mixture of barley and apples to dress wounds [114]. Although they did not understand the mode of action of their remedies, healers were using fungal-derived antimicrobial compounds to prevent and combat any potential wound infections. However, the later identification of these compounds paved the way for major advances in therapeutic medicine by allowing for the scaling-up of their production and increasing the diversity of active compounds through the use of semi-synthetic chemistry. The discovery that a myriad of bioactive molecules could be found in nature acted as a catalyst in research efforts to study fungal organisms and identify new bioactive compounds.

However, not all the compounds that exhibit interesting biological properties are made into active pharmaceutical ingredients. Although it is difficult to estimate the exact numbers, a rough approximation is that, for every new drug being released onto the market, at least 10,000 compounds showing potential biological activity were screened [115]. The bioactivities of NPs are always first tested only against specific fungal or bacterial strains, or against specific cancer lines in laboratory conditions; however, due to their high toxicity, many of these molecules also affect non-target cells, rendering the compound unsuitable for medicinal use. Although highly toxic NPs are unlikely to reach the pharmaceutical market, the diversity of their chemical structures can serve as a source of inspiration for chemists to synthesise molecules with a more targeted activity.

Another potential barrier to commercialisation is the ability to increase production of bioactive NPs. Secondary metabolites are generally synthesised by fungi in response to specific environmental conditions, which are difficult to reproduce artificially. Furthermore, these metabolites are often synthesised in small quantities. However, several methods of strain improvement have been used to increase the ability of a fungus to consistently produce NPs in large amounts. For instance, untargeted mutagenesis and alteration of the growing medium has increased the production of pneumocandins in *G. lozoyensis* by over 100% and 300%, respectively [116]. In recent years, synthetic biology methods, such as the heterologous expression of fungal biosynthetic gene clusters and targeted mutagenesis, have become prominent methods to significantly increase metabolite production. The antibiotic pleuromutilin was originally isolated from the basidiomycete *Clitopilus passeckerianus* [117]. The reconstruction of the pleuromutilin biosynthetic gene cluster in an engineered strain of *A. oryzae* resulted in over 1000% increase of antibiotic titration [118]. To support the production of medicinal drugs derived from fungal NPs, commercial-scale synthesis must be achieved, potentially through optimising fungal strains, growth conditions and incorporating techniques, such as pathway engineering.

The present review mostly focused on antimicrobial properties of fungal NPs, but the value of these secondary metabolites extends far above that of the antibiotic activity. For instance, placlitaxel (commercialised under the named Taxol^®^), which is one of the most-used anticancer drugs, is a fungal NP originally isolated form *Taxomyces andreanae*, which is an endophyte of the pacific yew tree (*Taxus brevifolia*) [119]. Similarly, statins, such as lovastatin produced by *A. terreus*, are fungal NPs with hypolipidemic properties that inhibit the 3-hydroxy-3-methylglutaryl-CoA reductase, an enzyme involved in cholesterol biosynthesis [120]. Statins are, at present, the most important class of cholesterol-lowering pharmaceuticals, further highlighting the importance of fungal NPs. Considering that, (i) relative to the large diversity of fungal species, only a handful of fungi have been studied and cultivated and, (ii) based on genome mining studies, each fungus probably has the ability to produce several NPs, after years of research into fungal NPs, humankind has only started to scratch the very surface of this diverse world. Therefore, the vast majority of bioactive compounds remain yet to be identified, and perhaps within these molecules lie the foundations of the next pharmaceutical drug families.

## Figures and Tables

**Figure 1 molecules-25-00911-f001:**
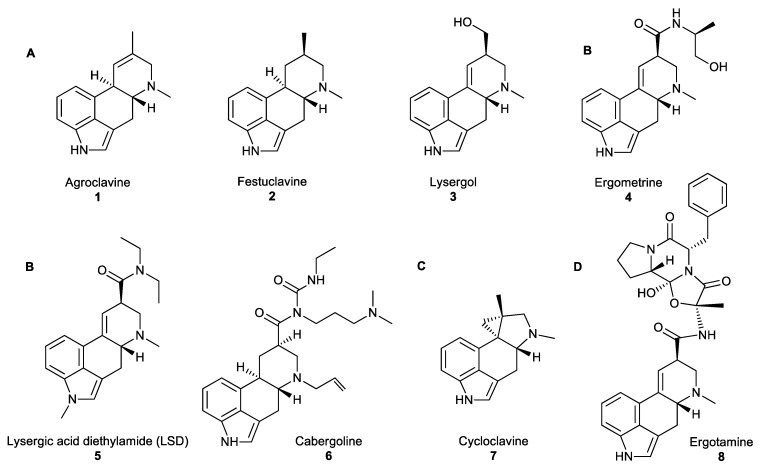
(**A**) Examples of clavines. (**B**) Simple lysergic acid derivatives. (**C**) The unusual ergoline scaffold of cycloclavine. (**D**) Ergopeptides consisting of D-lysergic acid with a cyclic tripeptide moiety [14].

**Figure 2 molecules-25-00911-f002:**
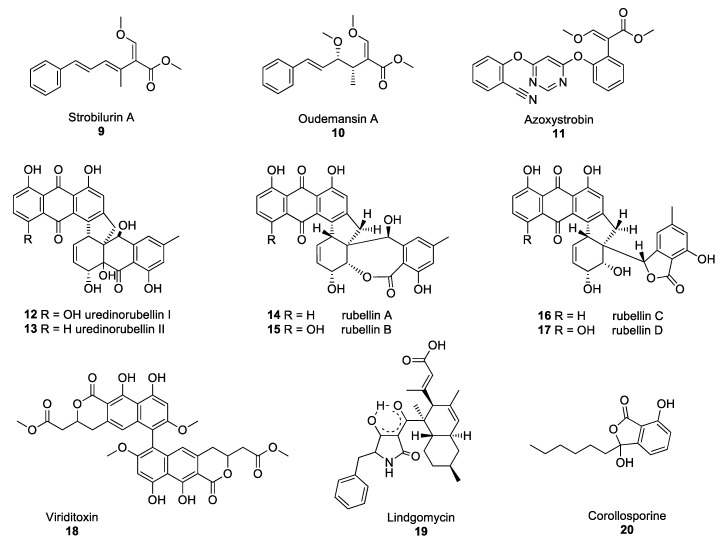
Structures of the selected fungal polyketides: Strobilurin A **9**, oudemansin A **10**, azoxystrobin **11**, uredinorubellins I and II (**12**–**13**) rubellins A–D (**14**–**17**), viriditoxin **18**, lindgomycin **19**, corollosporine **20**.

**Figure 3 molecules-25-00911-f003:**
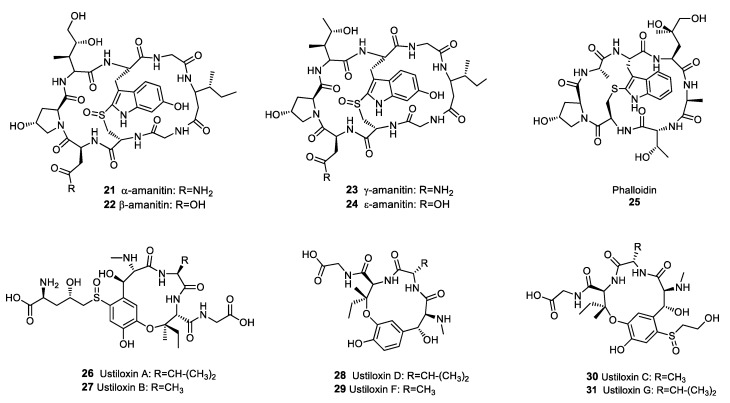
Structures of selected fungal ribosomally synthesised and post-translationally modified peptides (RiPPs).

**Figure 4 molecules-25-00911-f004:**
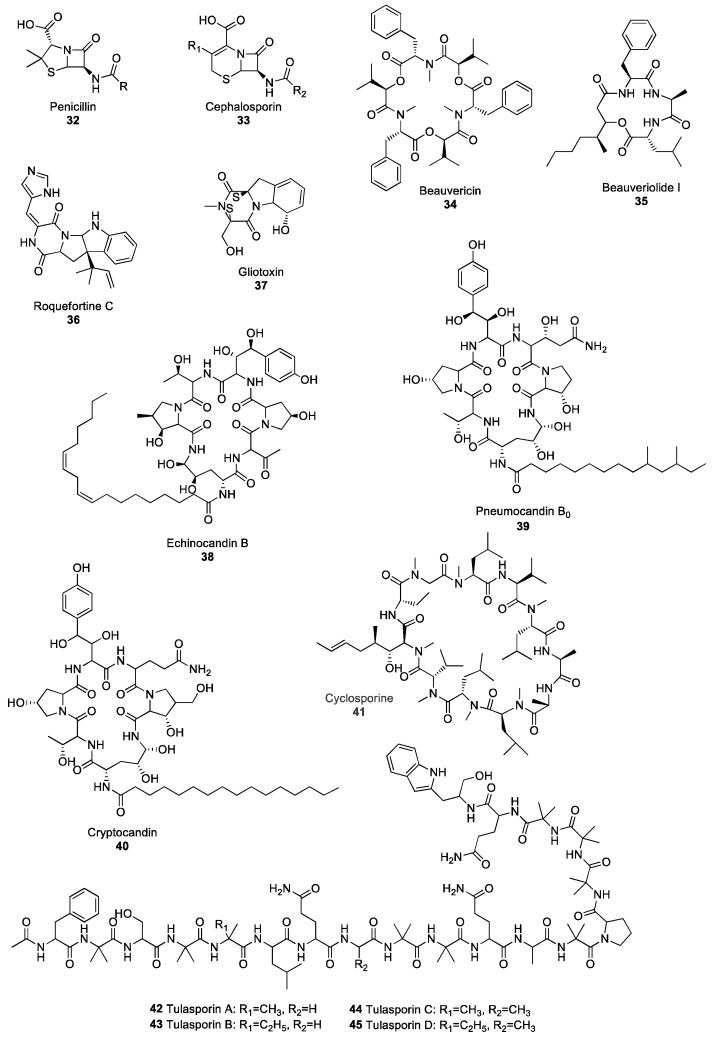
Diverse structures of selected non-ribosomal peptide (NRP)-derived fungal natural products.

**Figure 5 molecules-25-00911-f005:**
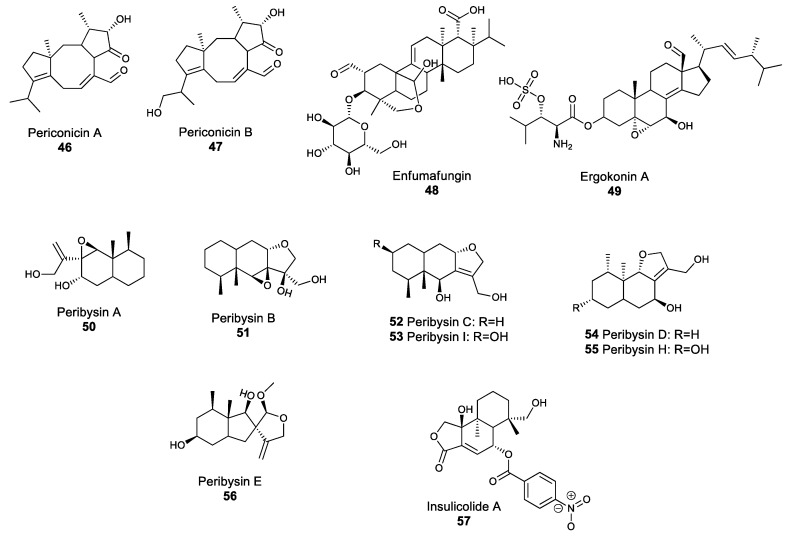
Structures of selected fungal terpenoids.

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
