# Peer review of "Selected Fungal Natural Products with Antimicrobial Properties"

_molecules, 2020, doi:10.3390/molecules25040911_

Round 1

Reviewer 1 Report

The review carried out by Jakubczyk and Dussart proved to be very comprehensive with varied and pertinent references. So I indicate the paper for publication.

Author Response

We would like to thank the reviewers for their comments. We have amended the manuscript following the reviewer's suggestion and appreciate the help of the reviewers in perfecting our manuscript.

Reviewer 1.

Reviewer comment: The review carried out by Jakubczyk and Dussart proved to be very comprehensive with varied and pertinent references. So I indicate the paper for publication.

Response: Many thanks for appreciating our work.

Reviewer 2 Report

The manuscript presented by Jakubczyk and Dussart describes the history of research on fungal natural products highlighting several known compound classes. The manuscript is excellently written and structured and for each compound class information about the origin, biosynthesis, molecular structure and bioactivities is given. To my mind, the introduction is slightly overstated (“exceptionally interesting”, “very underrepresented, “exceptional place”), however I leave the decision whether to change or not to the authors. The manuscript should definitely be considered for publication in MDPI Molecules and I see only little room for improvement. The only aspect that might be included comprises the source of the compounds commercially available. Are such compounds isolated from the native producer (or engineered native producer), synthesized chemically or biotechnologically produced by heterologous microorganism? For addressing this, a short section can be included in the conclusion section as it is relatively short in the current version of the manuscript.

Please also revise the manuscript based on a smaller number of minor comments given below:

line 27: Use of “NPs” instead of “NP” as abbreviation facilitates reading of the text. Check entire manuscript. 33: “EAs” 38: “Ergotismus gangraenosus” should not be written in italics as it might be misinterpreted as an organism name. 43: “oxytocic” is technical term a broader range of reader might not be familiar with. Consider to explain briefly. 55: “The next …” If possible, try to find a better connection to the preceding paragraph. In the current version it is very enumerative. 57: “To date, ….” 59: “reached the world market as antimicrobial compounds and are used against pathogenic bacteria”. This is basically the same. When they reach the market they are also applied?! 60: It is not clear what “they” refers to as “pathogenic bacteria” were the last mentioned term. 67: Full stop is missing at the end of the sentence. 85: The abbreviation was already introduced before. 100: add comma after “Eurotiales” 105: add comma after “elsewhere” 108: “N” in italics (appears twice) 136: “β-keto groups” 137: This is not necessarily true. Type III PKSs are rather small single-domain proteins (3oo-400 aa) also found in fungi. You refer to type I PKS proteins. 146: “tenacellus” 157: “point mutation …. leads to the substitution ….”. A mutation can only occur in genes, amino acids substitution in proteins. 185/216: The abbreviation NRPS should be introduced when mentioned for the first time. 185: “co-expression of genes coding for …” (protein cannot be expressed) 190-92: only the species name, not the strain name of deposition number should be written in italics. 212: “near the island Heligoland, Germany” 247: “β-lactam”, check entire manuscript, in the section title “beta” is used. 253: “synthetase 266: Consider to mention β-lactamases as resistance genes to this class of antibiotics, which are distributed amongst different bacterial phyla. The respective genes can be transferred to sensitive species via horizontal gene transfer, which then acquire resistance. 279: remove comma after “including” 314: “Aspergillus species” (italics) 319: “echinocandin” 342: “synthetase” 368: add comma after NP 420: “started to scratch”

Author Response

We would like to thank the reviewers for their comments. We have amended the manuscript following the reviewer's suggestion and appreciate the help of the reviewers in perfecting our manuscript.

Reviewer 2.

Reviewer comment: To my mind, the introduction is slightly overstated (“exceptionally interesting”, “very underrepresented, “exceptional place”), however, I leave the decision whether to change or not to the authors.

Response: The introduction has been revised to address this comment.

Reviewer comment: The only aspect that might be included comprises the source of the compounds commercially available. Are such compounds isolated from the native producer (or engineered native producer), synthesized chemically or biotechnologically produced by heterologous microorganism?

Response: A section discussing the importance of strain improvement and heterologous expression has been added in the concluding remark as suggested.

Reviewer comment: Please also revise the manuscript based on a smaller number of minor comments given below:

Response: The manuscript has been modified following all of the comments below

line 27: Use of “NPs” instead of “NP” as abbreviation facilitates reading of the text. Check entire manuscript. The manuscript has been modified throughout to address this comment.

33: “EAs” This has been changed in the text

38: “Ergotismus gangraenosus” should not be written in italics as it might be misinterpreted as an organism name. The sentence has been amended to avoid this possible confusion

43: “oxytocic” is technical term a broader range of reader might not be familiar with. Consider to explain briefly. A brief definition has been added to the text.

55: “The next …” If possible, try to find a better connection to the preceding paragraph. In the current version it is very enumerative. The sentence has been amended to address this comment

57: “To date, ….” A comma has been added

59: “reached the world market as antimicrobial compounds and are used against pathogenic bacteria”. This is basically the same. When they reach the market they are also applied?! “and are used against pathogenic bacteria” has been deleted to avoid repetition and confusion

60: It is not clear what “they” refers to as “pathogenic bacteria” were the last mentioned term. The pronoun has been replaced to avoid potential confusion

67: Full stop is missing at the end of the sentence. Full stop was added

85: The abbreviation was already introduced before. This has been addressed and the manuscript has been checked for other occurrences

100: add comma after “Eurotiales”  A comma was added

105: add comma after “elsewhere”. A semi-colon was added after NP

108: “N” in italics (appears twice)  The font has been changed for both occurrences

136: “β-keto groups” Ketone has been replaced by keto

137: This is not necessarily true. Type III PKSs are rather small single-domain proteins (3oo-400 aa) also found in fungi. You refer to type I PKS proteins. The sentence has been amended to account for this remark: the terms “huge” and “multimodular” have been removed.

146: “tenacellus” The spelling mistake has been corrected

157: “point mutation …. leads to the substitution ….”. A mutation can only occur in genes, amino acids substitution in proteins. The sentence was also split into 2 shorter sentences

185/216: The abbreviation NRPS should be introduced when mentioned for the first time. The abbreviation was introduced now on line 185

185: “co-expression of genes coding for …” (protein cannot be expressed) The sentence has been modified to address this comment

190-92: only the species name, not the strain name of deposition number should be written in italics. This has been addressed and the manuscript has been checked to improve consistency in using an abbreviation of species names after the first mention of binomial, genus names were abbreviated using the first letter capitalised. (e.g. Z. tritici)

212: “near the island Heligoland, Germany” This spelling mistake has been corrected

247: “β-lactam”, check entire manuscript, in the section title “beta” is used. The title has been changed to β for consistency

253: “synthetase  The spelling mistake has been corrected

266: Consider to mention β-lactamases as resistance genes to this class of antibiotics, which are distributed amongst different bacterial phyla. The respective genes can be transferred to sensitive species via horizontal gene transfer, which then acquire resistance. Few sentences have been added to address this comment.

279: remove comma after “including” Comma has been removed

314: “Aspergillus species” (italics) This comment has been addressed

319: “echinocandin” The spelling mistake was corrected

342: “synthetase” The spelling mistake was corrected

368: add comma after NP A semi-colon was added after NP

420: “started to scratch” The spelling mistake was corrected